# Association of intrauterine synechiae with pituitary gonadotrophin pulse patterns: A pilot study

**Arlete Gianfaldoni, Cristiane Roa, Ricardo dos Santos Simões, Maria Cândida P. Baracat, Angela Maggio da Fonseca, Vicente Renato Bagnoli, Isabel Cristina Espósito Sopreso, Fernando Wladimir Silva Rivas, Pedro Monteleone, Edmund C. Baracat, José Maria Soares Júnior***

Disciplina de Ginecologia, do Departamento de Obstetrícia e Ginecologia, Hospital das Clínicas, Faculdade de Medicina da Universidade de São Paulo, São Paulo, São Paulo, Brazil

* jsoares415@hotmail.com

## Abstract

### Background

Intrauterine synechiae (IS) is an acquired uterine condition that occurs when scar tissues (adhesions) form within the uterus and/or cervix, causing menstrual disturbance. However, approximately 50% of patients with IS are refractory to treatment. Therefore, other endocrine disturbances, such as gonadotropin disturbance, may affect treatment success.

### Study aim

To analyze gonadotropin levels in women with and without IS.

### Methods

Ten women with refractory IS experiencing amenorrhea since at least 6 months and nine with normal menstrual cycles (control group) were included in this study. Blood sample were collected every 10 minutes during a 4-h period. The serial ultrasound was performed in both groups for evaluating the cycle phase. Blood was collected when the follicles size was between 5–10 mm. Serum LH, FSH, progesterone and estradiol concentrations were measured. To detect LH and FSH pulses, the technique proposed by Santen and Bardin was adopted; therefore, one pulse was defined as a 20% increase in the concentrations as to the preceding point, followed by an important decrease.

### Results

No differences were observed between the study groups at baseline. Estradiol levels were lower in the IS group than in the control group, but the difference was not statistically significant. During the first hour of monitoring, cumulative FSH pulsatile frequency of IS group was lower than one of control.

**Data Availability Statement:** All relevant data are within the manuscript and its Supporting information files.

**Funding:** This study was funded by Cnpq, Brasilia-Br (#304264/2021-0) AND CAPES (#88887.621109/2021-00). The funders had no role in study design, data collection and analysis, decision to publish, or preparation of the manuscript.

**Competing interests:** The authors have declared that no competing interests exist.

## Conclusion

Our data suggest that the estradiol levels of IS participants are lower than those of women with normal menstrual cycle. The role of this finding in the physiology of uterine synechiae requires further investigation.

## Introduction

Intrauterine adhesions that develop as a result of trauma to the endometrial cavity, also known as intrauterine synechiae (IS), are usually preceded by puerperal or post-abortion curettage [1–3].

Many women have menstrual disorders (hypomenorrhea, oligomenorrhea, hypo-oligomenorrhea, amenorrhea, etc.), sterility, miscarriage, premature birth, and rarely, placental accretism and placenta previa [4], of which some are asymptomatic [5–7].

Conditions that allow the formation of intrauterine adhesions generally occur in the first 30 days of puerperium; the most important of which is drinking alcohol during pregnancy, which can make the uterus more prone to deep injury by curette, and postpartum hypogonadism, which can delay endometrial epithelialization. Under normal conditions, this would prevent adhesions between the internal surfaces of the uterus [8–11].

Previous studies have reported that hormonal cycle and ovulation are not altered in patients with traumatic intrauterine adhesions [1–3, 12–14]. Serum LH and FSH levels have been shown to be within normal limits in these patients [15]. However, functional changes in the hypothalamic-pituitary system of patients with IS were reported by Teter et al. [16]. In 1987, Halbe et al. [17] suggested neural involvement to explain the finding of an LH/FSH ratio greater than two in patients with canalicular amenorrhea.

Kheifets and Burtchik, 1984 [18], observed alterations in the hypothalamic-pituitary-ovarian system that were related to duration, extent, and location of intrauterine adhesions. Menstrual disorders have suggested changes in the pulsatility of pituitary gonadotropins, and it is not always the serum hormone concentration that defines normality or abnormality, but how these concentrations are maintained [19].

To date, the reduced number of studies related to neuroendocrine alterations and traumatic intrauterine adhesions, as well as lack of published studies on pituitary gonadotropin pulsatility in patients with menstrual alterations related to IS, motivated us to carry out clinical research to observe the pulsatility of LH and FSH in these patients, comparing it with that of normal women.

Macrophages are involved in endometrial repair and fibrosis. Studies have shown a decrease in endometrial macrophages and reduced expression of CSF1 (colony-stimulating factor 1) in women with IS [20]. These authors also suggested that there could be complications on the reproductive axis, resulting from this lower production. In female mouse models, CSF1 is known to modify the hypothalamic response to estrogen [21]. However, the actual impact of pituitary gonadotropins on pulsatility is unknown. Therefore, this study aimed to assess whether women with IS could have a second factor of infertility due to changes in the production of gonadotropins. Therefore, we reassessed patients seen in the Human Reproduction Service of the Gynecological Clinic of the Department of Obstetrics and Gynecology of the Faculty of Medicine of the University of São Paulo.

## Methods

### Eligibility criteria

We included women aged over 18 years with complaints of infertility who were treated in the Human Reproduction Service of the Gynecological Clinic of the Department of Obstetrics and

Gynecology of the Faculty of Medicine of the University of São Paulo. Uterine synechiae was present in the cases group (G1) experiencing amenorrhea for at least six months before the study admission. In the control group (G2), women with male factor infertility had no cause of infertility or irregular cycles.

We excluded women with uterine malformation or sexual development disorders, amenorrhea after infectious diseases (tuberculosis), morbid obesity, metabolic syndrome, diabetes mellitus, premature ovarian failure, and systemic diseases (systemic lupus erythematosus, rheumatoid arthritis, cancer of any origin, diabetes mellitus, and systemic arterial hypertension). Patients who were using the following methods for less than six months were excluded: levonorgestrel-releasing intrauterine device, medications or hormonal contraceptives, tubal ligation, or use of herbal substances.

This exploratory retrospective case series study assessed 10 women diagnosed with traumatic intrauterine adhesion (IS), and 9 with regular menstrual cycles with male factor infertility as a control group. All the patients gave their clinical history and underwent gynecological examination. The diagnosis of intrauterine adhesions was confirmed using hysterosalpingography and diagnostic hysteroscopy.

Biochemical measurements were performed: thyroid stimulating hormone, free T4, prolactin, FSH, LH, estradiol, total and free testosterone, 17-OH progesterone, progesterone glycemic curve with 75 g of glucose, fasting glucose in the first phase of the menstrual cycle in women in the control group, and on any day in women with amenorrhea. In the control group, women with the identification of any alteration or presence of systemic disease were excluded. The same exclusion criteria were used for the cases group (G1).

The study was approved by the ethics committee of the Faculty of Medicine of the USP (IRB #4.225.739—CAAE 20637019.4.0000.0068).

## Collection of blood samples

The serial ultrasound was performed in both groups for evaluating the cycle phase. Blood was collected when the follicles size was between 5–10 mm. The collection was done every 10 min for a period of 4 h at the outpatient clinic, which was part of the protocol for investigating fertility as used in other studies [22].

For the collection of blood samples, the following protocols were followed: a) patients were fasted and remained in the horizontal supine position; b) intravenous puncture was performed with a size-19 catheter, and the vein was maintenance with a saline solution drip (20 drops per minute); and c) 25 blood samples were collected from each woman, covering 4 hours.

The collections were performed between the second and fifth days of the menstrual cycle (follicular phase) in the G2 group. The G1 patients were followed by serial ultrasounds, and the ultrasound was performed in the next cycle when there were at least five small cystic images in one or both ovaries, which could suggest an early follicular phase.

The samples were sent to the Hormonal Dosage Laboratory at the Gynecological Clinic for determination of hormone concentrations. Serum concentrations of LH and FSH were measured in triplicate, using the LH (Fluorometric Enzyme Immunoassay) and FSH (Fluorometric Enzyme Immunoassay, Baxter Diagnostics Inc.) methods. The specifications are based on the manufacturer's guidelines. Plasma progesterone and estradiol concentrations were measured in triplicate by radioimmunoassay (solid phase), using the DPC method. Blood samples for these dosages were taken at zero and 240 min. Plasma progesterone levels in both groups were measured to detect any extemporaneous luteal phase. When serum progesterone concentrations were greater than 2.0 ng/mL, it was considered as the luteal phase, and these women were excluded from the series.

### Criteria for detecting the pulsatile pattern of LH and FSH

The technique proposed by Santen and Bardin was adopted for the detection of LH and FSH pulses, according to which pulse occurs whenever there is a 20% increase in the hormonal concentration at a single point in relation to the previous point, followed by a sharp decrease. The formula used to determine the mean time of occurrence of pulses was $T0 = (N \times t/Np)$, where $T0$ = time of occurrence, $N$ = number of patients, $t$ = observation period, and $Np$ = number of pulses in the group [23].

### Statistical analysis

A convenience sample was taken for this pilot study with the goal of recruiting 30 women. The data observed in this study were statistically analyzed using two different objectives. First, there were differences in the mean values of the serum concentrations of LH, FSH, progesterone, and estradiol between the two study groups. Statistical significance was determined using the Mann Whitney test for the median difference at a critical value level of $p < 0.05$. Second, we evaluated eventual differences in the pulsatile patterns of LH and FSH in women in the two groups. Summary tables of hourly pulse relative frequencies were created, and the data were subjected to the chi-square test, and a critical value of $p < 0.05$ was adopted.

## Results

Thirty participants were analyzed, but 11 were excluded: group 1; did not wish to participate (n = 6), and group 2; high prolactin level (n = 1), diabetes mellitus (n = 2), and did not wish to participate (n = 2). The ages of women in G1 ranged between 20–36 years, with a median of 30 years, while in G2 group, ages varied between 28–36 years, with a median age of 34 years. There was no statistically significant difference between the groups (p = 0.147). No differences were identified in the general and gynecological physical examinations. The period of amenorrhea for women in the G1 group ranged between 18–84 months, with a median of 30 months. The menstrual cycle duration in the G2 group ranged between 26–31 days, with a median of 28 days (Table 1).

The mean values of duplicate serum concentrations of LH, FSH, progesterone, and estradiol from the 19 blood samples of each woman in both groups, with the cumulative rates of detected pulses of LH and FSH were shown in boxplot graphics (Figs 1 and 2). The mean values of the baseline serum concentrations of LH, FSH, progesterone, and estradiol for each patient in the G1 and G2 groups are shown in Table 1. When we studied the mean values of the serum concentrations of LH, FSH, progesterone, and estradiol, we observed that there were no statistically significant differences between the mean values of serum concentrations of LH, FSH, and progesterone, or in the mean serum concentration levels of estradiol ($p < 0.05$).

Table 2 shows the hourly relative frequencies (%) of the LH and FSH pulses for each patient in the G1 and G2 groups. The absolute frequencies of the presence and/or absence of LH and FSH pulses were compared using the chi-squared test. In this table, we observe a higher frequency of LH pulses among patients with IS, but this is not statistically significant, while the frequency of FSH pulses is higher among women with normal menstrual cycle, but only statistically different when evaluating the first hour.

Regarding the mean time for pulses to occur, we found that the patients in group one had an LH pulse every 10 min, whereas the women in group two had an LH pulse every 180 min. Among the G1 patients, the mean time of occurrence of an FSH pulse was 20 h compared to approximately 7 h in the women in the G2 group, respectively.

**Table 1. Baseline characteristics of study subjects.**

|  | Medians (min-max) | | | p value* |
|---|---|---|---|---|
|  | IS (n = 10) | Controls (n = 9) | Both groups (n = 19) |  |
| Age (years) | 30 (20–36) | 34 (28–36) | 31 (20–36) | 0.175 |
| Amenorrhea (months) | 30 (18–84) |  |  |  |
| Menstrual interval (days) |  | 28 (26–32) |  |  |
| LH (nUI/ml) | 3.86 (1.65–6.34) | 4.15 (2.03–6.17) | 4.09 (1.65–6.34) | 0.369 |
| FSH (nUI/ml) | 5.3 (4.19–7.14) | 4.72 (2.77–7.51) | 5.13 (2.77–7.51) | 0.289 |
| Progesterone (ng/ml) | 0.34 (0.25–0.77) | 0.22 (0.1–0.42) | 0.32 (0.1–0.77) | 0.055 |
| Estradiol (pg/ml) | 39.6 (16.35–157) | 82 (23.05–186.5) | 50.97 (16.35–186.5) | 0.050 |

IS: Intrauterine Synechiae

LH: Luteinizing hormone

FSH: Follicle stimulating hormone

*P-value for Mann Whitney test

## Discussion

The influence of IS on the menstrual cycle has been extensively studied for a long time. The first study conducted by Burtchik and Taranina (1985) was an experimental model with rats [24]. They observed that endometrial lesions induce ovarian alterations soon after pregnancy. This model revealed a break in the estrous cycle, and the rats underwent cycle lengthening and anovulation reduction. In fact, endometrial inflammatory changes have been shown to release cytokines that interfere with the hypothalamic-ovarian axis (CSF-1). In animals, CSF-1 can

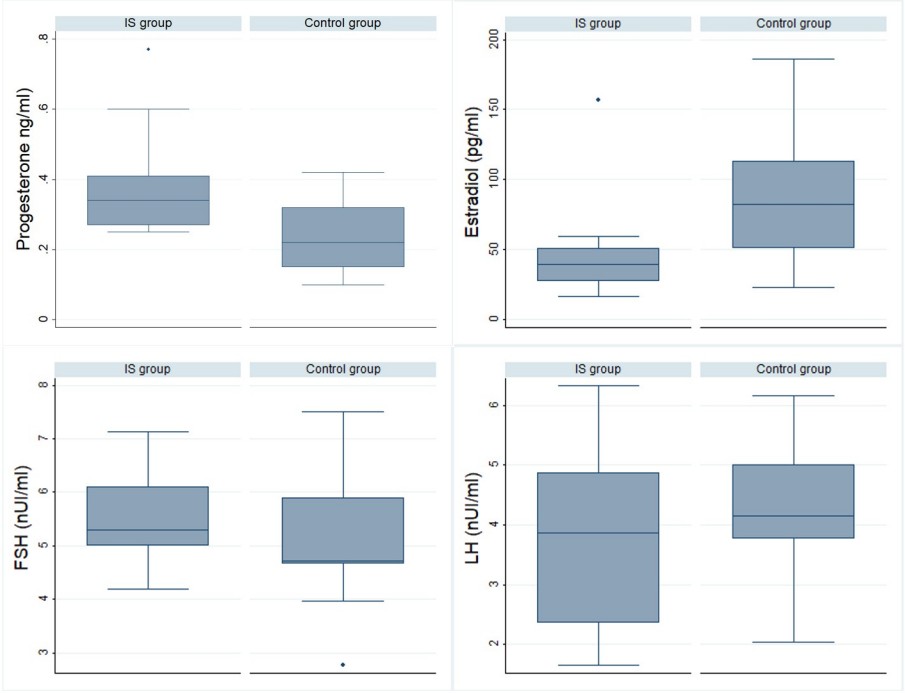

**Fig 1. Baseline progesterone, estradiol, FSH and LH levels.** IS: Intrauterine Synechiae, LH: Luteinizing hormone, FSH: Follicle stimulating hormone.

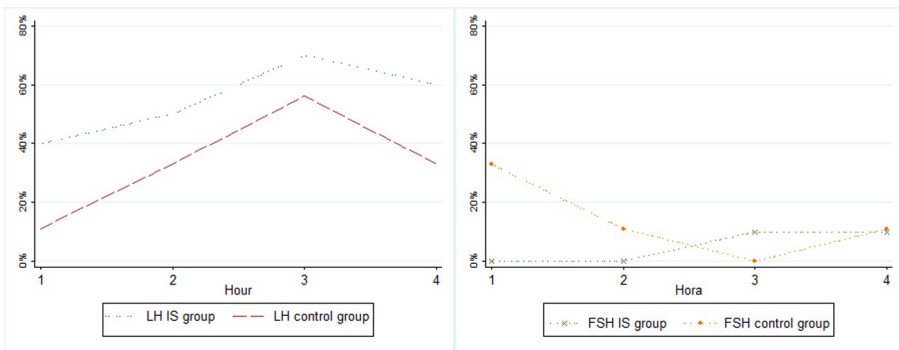

**Fig 2. Hourly cumulative FSH and LH pulses in intrauterine synechiae group and healthy controls.** IS: Intrauterine Synechiae, LH: Luteinizing hormone, FSH: Follicle stimulating hormone.

reduce LH levels [26]. Thus, the IS-related changes in our study could lead to a disturbance in the hypothalamic axis. Even after treatment, this can be another infertility factor for these women.

The influence of hormonal status at the time of injury are relevant factors on genesis [28]. Hypoestrogenism leading to reduced endometrial re-epithelialization facilitates the appearance of adhesions [12, 24–26]. Changes in ovarian function have been observed in the presence of intrauterine adhesion, leading to anovulation and hypoestrogenism [16, 18, 25–27]. However, we found that estradiol levels in the Asherrman participants were lower than those in the control patients. This may be an endocrine factor to consider after treating the patients with

**Table 2. Hourly cumulative frequency of LH and FSH pulses.**

|  |  | IS (n = 10) | Controls (n = 9) | Both groups (n = 19) | *p value** |
|---|---|---|---|---|---|
| **LH 1st hr.** | - | 60% | 89% | 73.68% | 0.153 |
|  | + | 40% | 11% | 26.32% |  |
| **LH 2nd hr.** | - | 50% | 67% | 57.89% | 0.463 |
|  | + | 50% | 33% | 42.11% |  |
| **LH 3rd hr.** | - | 30% | 44% | 36.84% | 0.515 |
|  | + | 70% | 56% | 63.16% |  |
| **LH 4th hr.** | - | 40% | 67% | 52.63% | 0.245 |
|  | + | 60% | 33% | 47.37% |  |
| **FSH 1st hr.** | - | 100% | 67% | 84.21% | **0.047** |
|  | + | 0% | 33% | 15.79% |  |
| **FSH 2nd hr.** | - | 100% | 89% | 94.74% | 0.279 |
|  | + | 0% | 11% | 5.26% |  |
| **FSH 3rd hr.** | - | 90% | 100% | 94.74% | 0.330 |
|  | + | 10% | 0% | 5.26% |  |
| **FSH 4th hr.** | - | 90% | 89% | 89.47% | 0.937 |
|  | + | 10% | 11% | 10.53% |  |

IS: Intrauterine Synechiae

LH: Luteinizing hormone

FSH: Follicle stimulating hormone

*P-value for Pearson chi-square test

IS. A specific clinical protocol may be necessary for estrogen treatment of these patients. Additionally, low estrogen levels may contribute to endometrial trauma and facilitate uterine synechiae.

This finding supports the hypothesis that long-term IS promotes ovarian failure. In our patients, the disease duration ranged from one and a half to seven years, with a mean of 3.4 years, and according to Kheǐfets et al. [18], these patients would be in the group in which hypoestrogenism could occur. The patients evaluated in this study were not separated according to the extent and time of evolution of adhesions, as this division would result in extremely small subgroups.

The LH and FSH concentrations in peripheral blood are not maintained constantly, but rather in a pulsatile manner; it can occur in women with normal and altered menstrual cycles, with changes in amplitude and frequency during the cycle [27–30]. The frequency of LH pulses increases during the early follicular phase, when estradiol and progesterone levels are low [27, 31]. According to Reame et al. [32], increase in pulsatile frequency of LH is important for stimulating follicular secretion of estradiol. Therefore, LH appears to arise from positive feedback of estradiol. Among women in the control group, the pulsatile LH frequency observed in the follicular phase was one pulse every 180 min. The researchers observed the occurrence of a pulse every 100–120 minutes in the follicular phase of normal women. In the literature, the pulsatility of pituitary gonadotropins varies from one pulse every 60 min to one pulse every 4–6 hours, with a higher frequency observed during the initial follicular phase, and a lower frequency during the luteal phase of the menstrual cycle [27, 30, 32–36]. Therefore, the LH pulsatile frequency recorded for our control group agreed with the literature. The frequency of FSH pulses in the control group was one pulse every approximately 7 h. Soules et al. reported the difficulty of quantifying the frequency of FSH pulses due to the low natural amplitude of its fluctuations [27]. Despite this difficulty, we observed a lower pulsatile frequency of FSH than LH, possibly due to the long half-life of FSH, which tends to obscure individual pulses.

Plasma FSH concentrations decreased by 50% in the follicular phase when estradiol levels increased. The FSH showed an inverse relationship with estradiol, and is low when estradiol concentrations are above 150 pg/mL. In our control group, estradiol levels close to those mentioned above were observed in one-third of the cases, which may explain the relatively low pulsatile frequency of FSH. In relation to patients with intrauterine adhesions, the frequency of LH pulsatility (one pulse every 110 min) was within the normal parameters mentioned above for women in the control group. In fact, there were no statistically significant differences in the LH pulsatile frequencies between patients with adhesions and those with normal menstrual cycle. The frequency of FSH pulses found in the patients with intrauterine adhesions was one pulse every 20 h, with no significant difference compared with the control group. In our study, perhaps the hypoestrogenism found in patients with intrauterine adhesions presented a statistically significant difference in relation to the control group, and may be one of the factors responsible for the change in frequency of FSH pulses.

Finally, we observed a difference in FSH cumulated pulsatility in the first hour (0% for IS versus 33% for controls). However, it was not possible to compare these data with information from other authors because, to date, there is no other work that relates pulsatility and IS among the sources consulted [36]. Thus, further clinical studies are required to investigate the underlying mechanisms.

This study has some limitations. Catheter dripped saline solution while collecting samples could affect our findings regarding hormone blood concentrations. Small sample size possibly hides moderate differences between groups that we expected to occur at baseline and follow up comparisons. Our study strength was to associate possible ovarian-pituitary-hypothalamus

axis disorder to the presence of intrauterine synechiae through the observation of gonadotrophins pulsatility patterns.

## Conclusion

Cumulative FSH pulsatile frequency was lower in the IS group during the first hour, increasing to control levels in the following hours. Our data suggest that baseline estradiol levels are lower in patients with IS than in those with normal menstrual cycles. The role of this finding in the physiology of uterine synechiae requires further investigation.

## Supporting information

**S1 File.**
(XLSX)

## Author Contributions

**Conceptualization:** Cristiane Roa, Ricardo dos Santos Simões, Maria Cândida P. Baracat, Angela Maggio da Fonseca, Vicente Renato Bagnoli, Isabel Cristina Espósito Sopreso, Pedro Monteleone, Edmund C. Baracat.

**Data curation:** Fernando Wladimir Silva Rivas.

**Formal analysis:** Fernando Wladimir Silva Rivas.

**Investigation:** Arlete Gianfaldoni, Cristiane Roa, Ricardo dos Santos Simões, Maria Cândida P. Baracat, Isabel Cristina Espósito Sopreso.

**Methodology:** Cristiane Roa, Fernando Wladimir Silva Rivas.

**Project administration:** José Maria Soares Júnior.

**Supervision:** José Maria Soares Júnior.

**Writing – original draft:** Arlete Gianfaldoni, Cristiane Roa, Ricardo dos Santos Simões, Maria Cândida P. Baracat, Fernando Wladimir Silva Rivas, José Maria Soares Júnior.

**Writing – review & editing:** Arlete Gianfaldoni, Cristiane Roa, Ricardo dos Santos Simões, Maria Cândida P. Baracat, Angela Maggio da Fonseca, Vicente Renato Bagnoli, Isabel Cristina Espósito Sopreso, Fernando Wladimir Silva Rivas, Pedro Monteleone, Edmund C. Baracat, José Maria Soares Júnior.

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
