## [Decision Letter · Decision Letter 0]

26 Dec 2022

PONE-D-22-33049Association of refractory Asherman syndrome with pituitary gonadotrophin pulse patterns: A pilot studyPLOS ONE

Dear Dr. Silva Rivas,

Thank you for submitting your manuscript to PLOS ONE. After careful consideration, we feel that it has merit but does not fully meet PLOS ONE’s publication criteria as it currently stands. Therefore, we invite you to submit a revised version of the manuscript that addresses the points raised during the review process.

We look forward to receiving your revised manuscript.

Kind regards,

Ahmed Mohamed Maged, MD

Academic Editor

PLOS ONE

Journal Requirements:

3. We noted in your submission details that a portion of your manuscript may have been presented or published elsewhere. [DETAILS AS NEEDED] Please clarify whether this [conference proceeding or publication] was peer-reviewed and formally published. If this work was previously peer-reviewed and published, in the cover letter please provide the reason that this work does not constitute dual publication and should be included in the current manuscript.

4. We noted in your submission details that a portion of your manuscript may have been presented or published elsewhere. [abstract, tables 1 & 2, figures 1 & 2. See annex posterAG.pdf] Please clarify whether this [conference proceeding or publication] was peer-reviewed and formally published. If this work was previously peer-reviewed and published, in the cover letter please provide the reason that this work does not constitute dual publication and should be included in the current manuscript.

Additional Editor Comments:

Please respond to all reviewers comments point by point

Reviewer 1

Thank you for the invitation for reviewing this interesting study , aiming for discussing association of refractory Asherman syndrome with pituitary gonadotrophin pulse Patterns and recommended that a specific clinical protocol may be necessary for estrogen treatment of these patients. I'm asking the authors for clarification of the next items.

1- In abstract sector: you mentioned this phrase (In both groups, blood was collected during the follicular phase of the menstrual cycle). You should revise and readjust to avoid misleading readers, as group one presented with amenorrhea.

2- In abstract sector: you mentioned (During the first hour of monitoring, cumulative FSH pulsatile frequency was lower in the AS group than in the control group within the next 3 h of measurements). Rewrite the phrase more clearly

3- In introduction section, defining Asherman syndrome has about 11 references, what is the rationale for this great number? Can we minimize especially no benefit from the repetition ?.

4- The study was retrospective :

In methodology section, the statement (These patients had amenorrhea for at least six months after admission to the study). May be understood. You mean before admission to the study?

Also, I just want to confirm that 8 patients refused to participate?

5- In methodology section, Clarify the meaning of (The same was done for women with Asherman syndrome, exception the analysis of gonadotropins and sex steroids, estrogen, and progesterone.).

6- You have a reference for this blood sampling protocol ?

(For the collection of blood samples, the following protocols were followed: a) patients were fasted and remained in the horizontal supine position; b) intravenous puncture was performed with a size-19 catheter, and the vein was maintenance with a saline solution drip (20 drops per minute); and c) 25 blood samples were collected from each woman)

7- In the phrase (We observed that there was no statistically significant difference between the mean values of serum concentrations of LH, FSH, and progesterone, or in the mean serum concentration levels of estradiol (p>0.05)).

The (p value) equal or more? confirm.

8- In discussion section: what you mean by (The influence of hormonal status at the time of injury and relevant factors on genesis). There are missing words? I don’t understand.

9- I discussion section and at the end of page 10 in (pdf copy) (They observed the occurrence of a pulse every 100-120 minutes in the follicular phase of normal women)

The word they refer to whom???

10- You abbreviated asherman to AS in introduction section , however , I noticed you mention it again with full . Revise that allover the article .

Reviewer 2

1. Why did you choose the term “Asherman syndrome”. It is well known that Asherman syndrome is intrauterine synechia following surgical evacuation. The term “intrauterine synechia” is a more accepted term.

2. In the introduction section, in the last paragraph “correct “his” to “this”.

3. You collected 25 samples from each patient, and this was approved by your IRB. What was your rational for this? I mean what is the benefit that the patient was told to expect? I would be grateful if you would share your IRB. Additionally maintaining the IV line with saline infusion, would this not alter your results? Please explain.

4. What is meant by “The G1 patients were followed by serial ultrasounds, and the ultrasound was performed in the next cycle when there were at least five small cystic images in one or both ovaries, which could suggest an early follicular phase”

5. The discussion is well written but please express your points of strength and limitation.

Reviewer 3

Although the concept is interesting and Asherman syndrome is not frequent BUT this does not justify the very low number of participants in this study

Asherman syndrome has classification of severity : it seems that the authors included only the severe cases with Amenorrhoea : pls explain why not cases with moderate IU adhesions

The following statement in the manuscript is very unusual in research design : "These patients had amenorrhea for at least six months after admission to the study." why to wait six month after recruitment !!!!!!

The authors failed to mention which day of cycle blood samples were taken : is it the same day for all the participants for example cycle day 11

The authors failed to correlate the level of hormones with the follicle size at the day of sampling : what if day of sampling was when follicle 11mm in Asherman cases and follicle 15mm in control group : Hormonal assays differe according to follicular activity

Additional comments to be clarified

Exclusion criteria what about women with endocrinological disorders especially PCOS women

How was the diagnosis of refractory AS established based on what criteria

Was the trial registered and what is the registration number

How was the study retrospective

Clarify the duration of fasting before sampling

Please discuss the effect of saline infused on the results o sampled blood especially its dilutional effect

More details about kits and its sensitivity

Define primary and secondary outcomes with references

Remove amenorrhea duration and menstrual cycle from table 1

In discussion

Physiological explanation of findings , strengths and limitations of the study should be added to discussion

Masterdata lack age

Reviewers' comments:

Reviewer's Responses to Questions

**Comments to the Author**

1. Is the manuscript technically sound, and do the data support the conclusions?

Reviewer #1: No

Reviewer #2: Yes

Reviewer #3: Yes

2. Has the statistical analysis been performed appropriately and rigorously? 

Reviewer #1: No

Reviewer #2: Yes

Reviewer #3: I Don't Know

3. Have the authors made all data underlying the findings in their manuscript fully available?

Reviewer #1: Yes

Reviewer #2: Yes

Reviewer #3: Yes

4. Is the manuscript presented in an intelligible fashion and written in standard English?

Reviewer #1: Yes

Reviewer #2: Yes

Reviewer #3: Yes

5. Review Comments to the Author

Reviewer #1: Although the concept is interesting and Asherman syndrome is not frequent BUT this does not justify the very low number of participants in this study

Asherman syndrome has classification of severity : it seems that the authors included only the severe cases with Amenorrhoea : pls explain why not cases with moderate IU adhesions

The following statement in the manuscript is very unusual in research design : "These patients had amenorrhea for at least six months after admission to the study." why to wait six month after recruitment !!!!!!

The authors failed to mention which day of cycle blood samples were taken : is it the same day for all the participants for example cycle day 11

The authors failed to correlate the level of hormones with the follicle size at the day of sampling : what if day of sampling was when follicle 11mm in Asherman cases and follicle 15mm in control group : Hormonal assays differe according to follicular activity

Reviewer #2: Thank you for the invitation for reviewing this interesting study , aiming for discussing association of refractory Asherman syndrome with pituitary gonadotrophin pulse Patterns and recommended that a specific clinical protocol may be necessary for estrogen treatment of these patients. I'm asking the authors for clarification of the next items.

1- In abstract sector: you mentioned this phrase (In both groups, blood was collected during the follicular phase of the menstrual cycle). You should revise and readjust to avoid misleading readers, as group one presented with amenorrhea.

2- In abstract sector: you mentioned (During the first hour of monitoring, cumulative FSH pulsatile frequency was lower in the AS group than in the control group within the next 3 h of measurements). Rewrite the phrase more clearly

3- In introduction section, defining Asherman syndrome has about 11 references, what is the rationale for this great number? Can we minimize especially no benefit from the repetition ?.

4- The study was retrospective :

In methodology section, the statement (These patients had amenorrhea for at least six months after admission to the study). May be understood. You mean before admission to the study?

Also, I just want to confirm that 8 patients refused to participate?

5- In methodology section, Clarify the meaning of (The same was done for women with Asherman syndrome, exception the analysis of gonadotropins and sex steroids, estrogen, and progesterone.).

6- You have a reference for this blood sampling protocol ?

(For the collection of blood samples, the following protocols were followed: a) patients were fasted and remained in the horizontal supine position; b) intravenous puncture was performed with a size-19 catheter, and the vein was maintenance with a saline solution drip (20 drops per minute); and c) 25 blood samples were collected from each woman)

7- In the phrase (We observed that there was no statistically significant difference between the mean values of serum concentrations of LH, FSH, and progesterone, or in the mean serum concentration levels of estradiol (p>0.05)).

The (p value) equal or more? confirm.

8- In discussion section: what you mean by (The influence of hormonal status at the time of injury and relevant factors on genesis). There are missing words? I don’t understand.

9- I discussion section and at the end of page 10 in (pdf copy) (They observed the occurrence of a pulse every 100-120 minutes in the follicular phase of normal women)

The word they refer to whom???

10- You abbreviated asherman to AS in introduction section , however , I noticed you mention it again with full . Revise that allover the article .

Reviewer #3: I have read with interest the manuscript titled : “Association of refractory Asherman syndrome with pituitary gonadotrophin pulse patterns: A pilot study”. I would like to thank the authors for the effort put until this work came out to light. I just have a few comments,

1. Why did you choose the term “Asherman syndrome”. It is well known that Asherman syndrome is intrauterine synechia following surgical evacuation. The term “intrauterine synechia” is a more accepted term.

2. In the introduction section, in the last paragraph “correct “his” to “this”.

3. You collected 25 samples from each patient, and this was approved by your IRB. What was your rational for this? I mean what is the benefit that the patient was told to expect? I would be grateful if you would share your IRB. Additionally maintaining the IV line with saline infusion, would this not alter your results? Please explain.

4. What is meant by “The G1 patients were followed by serial ultrasounds, and the ultrasound was performed in the next cycle when there were at least five small cystic images in one or both ovaries, which could suggest an early follicular phase”

5. The discussion is well written but please express your points of strength and limitation.

6. PLOS authors have the option to publish the peer review history of their article (what does this mean?). If published, this will include your full peer review and any attached files.

Reviewer #1: **Yes: **Hesham Al-Inany

Reviewer #2: No

Reviewer #3: **Yes: **Hisham Haggag. Prof Of Ob/gyn Cairo University.

---

## [Author Response · Author response to Decision Letter 0]

9 May 2023

São Paulo, April 5, 2023

Re: Submission of manuscript entitled “Association of refractory Asherman syndrome with pituitary gonadotrophin pulse patterns: A pilot study”

Dear Prof Editor,

We would like to resubmit our manuscript entitled “Association of Refractory Asherman syndrome with pituitary gonadotrophin pulse patterns: A pilot study,” after corrections.

Reviewer #1

1. In abstract sector: you mentioned this phrase (In both groups, blood was collected during the follicular phase of the menstrual cycle). You should revise and readjust to avoid misleading readers, as group one presented with amenorrhea.

A: The serial ultrasound was performed to evaluate the cycle. We included in abstract.

2. In abstract sector: you mentioned (During the first hour of monitoring, cumulative FSH pulsatile frequency was lower in the AS group than in the control group within the next 3 h of measurements). Rewrite the phrase more clearly 

A: We rewrote it

3. In introduction section, defining Asherman syndrome has about 11 references, what is the rationale for this great number? Can we minimize especially no benefit from the repetition?

A: We reduced the references.

4. -The study was retrospective: In methodology section, the statement (These patients had amenorrhea for at least six months after admission to the study). May be understood. You mean before admission to the study? Also, I just want to confirm that 8 patients refused to participate? 

A: We clarified on the manuscript. Eight patients refused to collect blood.

5. In methodology section, Clarify the meaning of (The same was done for women with Asherman syndrome, exception the analysis of gonadotropins and sex steroids, estrogen, and progesterone.). 

A: The formula used to determine the mean time of occurrence of pulses.

6. You have a reference for this blood sampling protocol? (For the collection of blood samples, the following protocols were followed: a) patients were fasted and remained in the horizontal supine position; b) intravenous puncture was performed with a size-19 catheter, and the vein was maintenance with a saline solution drip (20 drops per minute); and c) 25 blood samples were collected from each woman)

A: This protocol was used at the time this study was done. 

7. In the phrase (We observed that there was no statistically significant difference between the mean values of serum concentrations of LH, FSH, and progesterone, or in the mean serum concentration levels of estradiol (p>0.05) The (p value) equal or more? confirm.

A: P critical values was 0,05. Then, P-values accepted as statistically significant are lower than 0,05.

8. In discussion section: what you mean by (The influence of hormonal status at the time of injury and relevant factors on genesis). There are missing words? I don’t understand.

A: There was Mistyping in this phrase, instead of “and” we will use the word “are”.

9. I discussion section and at the end of page 10 in (pdf copy) (They observed the occurrence of a pulse every 100-120 minutes in the follicular phase of normal women). The word they refer to whom???

A: We rewrite this phrase, the word “they” is referring to the researchers (Reame et. al.)

10. You abbreviated Asherman to AS in introduction section, however, I noticed you mention it again with full . Revise that allover the article.

A: We revised and changed the term “Asherman syndrome” and replaced with “Intrauterine Synechiae” (IS) using only the abbreviation after the first use of the full term.

Reviewer #2

1. Why did you choose the term “Asherman syndrome”. It is well known that Asherman syndrome is intrauterine synechia following surgical evacuation. The term “intrauterine synechia” is a more accepted term.

A: We agree, Intrauterine Synechiae is a more accepted term, every time we used Asherman Syndrome we will change it for Intrauterine Synechiae

2. In the introduction section, in the last paragraph “correct “his” to “this”.

A: The correct word is “this”. Rewrote 

3. You collected 25 samples from each patient, and this was approved by your IRB. What was your rational for this? I mean what is the benefit that the patient was told to expect? I would be grateful if you would share your IRB. Additionally maintaining the IV line with saline infusion, would this not alter your results? Please explain.

A: At the time the research was carried out there was a protocol for investigating infertility In patients with intrauterine synechiae, involving also investigating possible Hypothalamic-pituitary-ovarian axis disorder through the gonadotrophins pulsatility study, during the first four hours every ten minutes. 

The benefit to patients was to detect possible disorders in the Hypothalamic-pituitary-ovarian axis.

Saline solution dripping aimed to maintain venous permeability in a minimum quantity in both groups indistinctly. We wrote a phrase explaining how this may be a limitation of our study.

IRB number is: IRB# 4225739-CAAE 20637019400000068

4. What is meant by “The G1 patients were followed by serial ultrasounds, and the ultrasound was performed in the next cycle when there were at least five small cystic images in one or both ovaries, which could suggest an early follicular phase”

A: As most of G1 patients presented amenorrhea, serial ultrasound could suggest an early follicular phase, so we can schedule sample collection in these patients.

5. The discussion is well written but please express your points of strength and limitation.

A: As strength we consider, the association of Hypothalamic-pituitary-ovarian axis disorder to the presence of intrauterine synechiae through the observations of gonadotrophins pulsatility. As limitations we considered, small sample size.

Reviewer #3

1. Although the concept is interesting and Asherman syndrome is not frequent BUT this does not justify the very low number of participants in this study. Asherman syndrome has classification of severity: it seems that the authors included only the severe cases with Amenorrhea: pls explain why not cases with moderate IU adhesions.

A: patients were women with severe intrauterine adherences, at the time these patients were followed at outpatient facilities and accepted to participate. We did not choose these patients nor excluded moderate IU adhesions.

2. The following statement in the manuscript is very unusual in research design: "These patients had amenorrhea for at least six months after admission to the study." why to wait six month after recruitment !!!!!!

A: We did not wait 6 months, when these women were recruited, they already presented 6 months or more amenorrhea.

3. The authors failed to mention which day of cycle blood samples were taken: is it the same day for all the participants for example cycle day 11. 

A: Its not the same day for all participants, it was almost impossible to determine the cycle day, especially for those with amenorrhea. In these patients we use ultrasound suggesting that the patients were in early follicular phase.

4. The authors failed to correlate the level of hormones with the follicle size at the day of sampling: what if day of sampling was when follicle 11mm in Asherman cases and follicle 15mm in control group: Hormonal assays differ according to follicular activity

A: Actually, this situation can be considered as a limitation of our study.

Additional comments to be clarified

5. Exclusion criteria what about women with endocrinological disorders especially PCOS women

A: Yes, we agree, PCOS should have been used as an exclusion criterion.

6. How was the diagnosis of refractory AS established based on what criteria

 A: Patients has intrauterine adhesions confirmed by hysterosalpingography and hysteroscopy

7. Was the trial registered and what is the registration number

How was the study retrospective

A: The study was not registered as a trial. O IRB Number IRB# 4225739-CAAE 20637019400000068.

The study design is not retrospective. The study was done prospectively from group definition and baseline measurements to 4 hours follow up. The study was done in 1993 and we reevaluated the data.

8. Clarify the duration of fasting before sampling

A: 8-12 hours 

9. Please discuss the effect of saline infused on the results of sampled blood especially its dilutional effect.

A: Saline solution dripping aimed to maintain vein permeability, it possible that this situation may interfere in results affecting both groups. 

10. More details about kits and its sensitivity

A: We used LH Fluorometric Enzyme immunoassay and FSH Fluorometric Enzyme immunoassay and FSH, Baxter Diagnostics Inc. 

We did not review its sensitivity.

11. Define primary and secondary outcomes with references

Remove amenorrhea duration and menstrual cycle from table 1

In discussion.

A: Our primary outcome was gonadotrophins pulsatility and it was defined by Root et. al. (reference number 24). Hormone levels related to IS were used as an outcome by several authors, take as example our bibliographic references: 1-4, 17-19, 20-23. 

We removed all amenorrhea from table 1.

12. Physiological explanation of findings, strengths and limitations of the study should be added to discussion. Master data lack age. 

A: Intrauterine synechiae development may promote a decrease in activity of uterine muscular wall, reducing the perfusion of sexual hormones at endometrial level contributing to endometrium atrophy.

We added to discussion a paragraph about limitations ands strengths of our study. Age is in database.

---

## [Decision Letter · Decision Letter 1]

30 May 2023

PONE-D-22-33049R1Association of intrauterine synechiae with pituitary gonadotrophin pulse patterns: A pilot studyPLOS ONE

Dear Dr. Silva Rivas,

Thank you for submitting your manuscript to PLOS ONE. After careful consideration, we feel that it has merit but does not fully meet PLOS ONE’s publication criteria as it currently stands. Therefore, we invite you to submit a revised version of the manuscript that addresses the points raised during the review process.

Please respond to all reviewers' comments

We look forward to receiving your revised manuscript.

Kind regards,

Ahmed Mohamed Maged, MD

Academic Editor

PLOS ONE

Additional Editor Comments:

Please respond to all reviewers' comments

Reviewers' comments:

Reviewer's Responses to Questions

**Comments to the Author**

1. If the authors have adequately addressed your comments raised in a previous round of review and you feel that this manuscript is now acceptable for publication, you may indicate that here to bypass the “Comments to the Author” section, enter your conflict of interest statement in the “Confidential to Editor” section, and submit your "Accept" recommendation.

Reviewer #1: All comments have been addressed

Reviewer #2: All comments have been addressed

2. Is the manuscript technically sound, and do the data support the conclusions?

Reviewer #1: Yes

Reviewer #2: Yes

3. Has the statistical analysis been performed appropriately and rigorously? 

Reviewer #1: Yes

Reviewer #2: Yes

4. Have the authors made all data underlying the findings in their manuscript fully available?

Reviewer #1: Yes

Reviewer #2: Yes

5. Is the manuscript presented in an intelligible fashion and written in standard English?

Reviewer #1: (No Response)

Reviewer #2: Yes

6. Review Comments to the Author

Reviewer #1: (No Response)

Reviewer #2: Dear editor ,

Thank you for your second invitation for revision evaluation and thanks for authors reply .

Actually, the study design (retrospective or prospective )should be clarified in the manuscript with the date of IRP approval

including patient refusal and the duration of amenorrhea before recruitment .

Additionally, make an effort to get details on the appropriate sampling technique to reduce errors as, from what I understand, it was remote .As an example, to ensure for a trial to proceed properly, blood including fluid was aspirated and drawn before the sample was obtained and submitted for analysis and keep that also in the limitation.

7. PLOS authors have the option to publish the peer review history of their article (what does this mean?). If published, this will include your full peer review and any attached files.

Reviewer #1: **Yes: **Hesham Al-Inany

Reviewer #2: **Yes: **Essamedin M Negm

---

## [Author Response · Author response to Decision Letter 1]

7 Jul 2023

Sao Paulo, July 5, 2023

Dear Plos One editors and reviewers

We would like to submit our manuscript entitled “Association of intrauterine synechiae with pituitary gonadotrophin pulse patterns: A pilot study”. We already rewrote several term as suggested by Reviewer # 1. Now we send you this corrected version including the Reviewer#2 suggestions. All authors agree with this version, we hope to meet the required changes and to clarify your questions about our work.

1- In abstract sector: you mentioned this phrase (In both groups, blood was collected during the follicular phase of the menstrual cycle). You should revise and readjust to avoid misleading readers, as group one presented with amenorrhea.

A1: Blood sample were collected every 10 minutes during a 4-h period. The serial ultrasound was performed in both groups for evaluating the cycle phase. Blood was collected when the follicles size was between 5-10 mm.

2- In abstract sector: you mentioned (During the first hour of monitoring, cumulative FSH pulsatile frequency was lower in the AS group than in the control group within the next 3 h of measurements). Rewrite the phrase more clearly 

A2: We changed the phrase to: “During the first hour of monitoring, cumulative FSH pulsatile frequency was lower in the IS group than in the control group”.

3- In introduction section, defining Asherman syndrome has about 11 references, what is the rationale for this great number? Can we minimize especially no benefit from the repetition?

A3: We checked and removed 5 references.

4- The study was retrospective: 

In methodology section, the statement (These patients had amenorrhea for at least six months after admission to the study). May be understood. You mean before admission to the study?

Also, I just want to confirm that 8 patients refused to participate? 

A4: We mean “before admission”, we rewrote the phrase:

“Uterine synechiae was present in the cases group (G1) experiencing amenorrhea for at least six months before the study admission”.

Yes, 6 patients in the cases group (G1) and 2 in the control group (G2) refuse to participate in the study, refused to participate when evaluated in the outpatient endocrine gynecologic clinic.

5- In methodology section, Clarify the meaning of (The same was done for women with Asherman syndrome, exception the analysis of gonadotropins and sex steroids, estrogen, and progesterone.). 

A5: We wrote incorrectly the phrase: we meant “the same exclusion criteria were used for the cases group (G1)” 

(Page 4)

6- You have a reference for this blood sampling protocol?

The protocol was done following the methods used in reference number 22:

• Fenichel RM, Dominguez JE, Mayer L, Walsh BT, Boozer C, Warren MP. Leptin levels and luteinizing hormone pulsatility in normal cycling women and their relationship to daily changes in metabolic rate. Fertil Steril. 2008 Oct;90(4):1161-8

7- In the phrase (We observed that there was no statistically significant difference between the mean values of serum concentrations of LH, FSH, and progesterone, or in the mean serum concentration levels of estradiol (p>0.05)).

The (p value) equal or more? confirm.

A7: The p value used to define statistical differences was less than 0.05, we already fix this typing error (Page 7)

8- In discussion section: what you mean by (The influence of hormonal status at the time of injury and relevant factors on genesis). There are missing words? I don’t understand.

We replaced with: “The influence of hormonal status at the time of injury are relevant factors on genesis (28)”

(Page 7)

9- I discussion section and at the end of page 10 in (pdf copy) (They observed the occurrence of a pulse every 100-120 minutes in the follicular phase of normal women)

The word they refer to whom???

A9: We are referring to the researchers of the cited study. 

We rewrote: “The researchers observed the occurrence of a pulse every 100-120 minutes in the follicular phase of normal women”

(page 8)

10- You abbreviated Asherman to AS in introduction section, however, I noticed you mention it again with full. Revise that allover the article.

A10: We already replaced the term Asherman syndrome to Intrauterine synechiae and used the abbreviation “IS”.

---

## [Editor Report · Decision Letter 2]

11 Jul 2023

Association of intrauterine synechiae with pituitary gonadotrophin pulse patterns: A pilot study

PONE-D-22-33049R2

Dear Dr. Silva Rivas,

We’re pleased to inform you that your manuscript has been judged scientifically suitable for publication and will be formally accepted for publication once it meets all outstanding technical requirements.

Kind regards,

Ahmed Mohamed Maged, MD

Academic Editor

PLOS ONE
---

## [Editor Report · Acceptance letter]

27 Jul 2023

PONE-D-22-33049R2 

Association of intrauterine synechiae with pituitary gonadotrophin pulse patterns: A pilot study 

Dear Dr. Silva Rivas:

I'm pleased to inform you that your manuscript has been deemed suitable for publication in PLOS ONE. Congratulations! Your manuscript is now with our production department. 

Kind regards, 

on behalf of

Professor Ahmed Mohamed Maged 

Academic Editor

PLOS ONE